# Diagnosis, Treatment and Prognosis of Primary Pulmonary NUT Carcinoma: A Literature Review

**Jiaqian Yuan** [1] , **Zhili Xu** [2] **and Yong Guo** [2,*]

1 The First Clinical Medical College, Zhejiang Chinese Medical University, Hangzhou 310053, China
2 Department of Medical Oncology, The First Affiliated Hospital of Zhejiang Chinese Medical University, Hangzhou 310001, China
* Correspondence: guoyong1047@gmail.com

**Abstract:** NUT carcinoma is a rare, highly lethal cancer characterized with the rearrangement of the nuclear protein in testis (NUT) gene on chromosome 15q14, which primarily occurs in the midline organs. Primary pulmonary NUT carcinoma (NC) lacks characteristic clinical manifestations, which leads to the high rate of misdiagnose and nonstandard treatment. To date, fewer than one hundred cases have been reported worldwide. Here, a comprehensive literature search involving a total of 35 articles with 55 patients was conducted in this paper. We reviewed and analyzed the associated clinical and pathological characteristics, the efficacy of various treatment options and the prognosis. Pulmonary NC mainly occurred in middle young-aged men (median age, 36) with no smoking history (2:1) and would present with symptoms of cough (63.6%), dyspnea (29.5%), chest pain (18.2%) and hemoptysis (18.2%). The initial imaging frequently revealed large and irregular lesions in the lower lobe (46.5%) of the left or right lungs; lymph node metastasis was also prevalent (91.9%). A focal squamous differentiation with abrupt keratinization often occurred in the undifferentiated or poorly differentiated (93.75%) tumor cells, with abundant necrosis and numerous neutrophils infiltrated. The mean overall survival (OS) in patients of this malignant disease was 6.21 months, and the median OS was 4.4 months. According to our results, this disease is sensitive to radiotherapy, and chemoradiotherapy (either concurrent chemoradiotherapy or sequential chemoradiotherapy) was the most efficient therapeutic regimen to prolong the OS of patients with pulmonary NC.

**Keywords:** NUT carcinoma (NC); primary pulmonary NC; chemoradiotherapy

## 1. Introduction

Nuclear protein of the testis (NUT) is a protein translated by the NUTM1 gene (NUT family member 1, located on the long arm of chromosome 15) whose expression should be restricted to the testis and ovary. Nuclear protein of the testis carcinoma (NC), also known as NUT midline carcinoma (NMC), is a rare and highly aggressive malignant epithelial tumor that typically affects midline organs, including the nasal cavity, palate and mediastinum. The pathogenesis of NC is currently unclear, which is often thought to be related to the NUT-mediated genome-wide histone modification that alters the expression of oncogenes or tumor suppressor genes. Molecular genetic studies have demonstrated recurrent fusions involving the NUT gene on chromosome 15q14 [1]. Related cases have reported fusions involve BRD4 on chromosome 19p13.1. Frequently, BRD3 and NSD3 have also been reported. The prognosis of NC is often extremely poor. A report showed [2] that the disease progresses rapidly to death in most cases, with a median overall survival (OS) of only 6.7 months and a 2-year progression-free survival (PFS) of 9%.

The World Health Organization (WHO, Geneva, Switzerland) included primary pulmonary NC in the classification of other undifferentiated cancers of lung malignancies in 2015. However, less than 100 cases of primary pulmonary NC have been reported worldwide. Due to the rarity, the occurrence of the primary pulmonary NC is often ignored

by clinicians, and its clinical understanding is not yet profound. In this paper, the relevant clinical characteristics, imaging and pathological features of primary pulmonary NC, as well as the treatment methods and prognosis for a total of 55 patients, were summarized from 35 articles in order to deepen the understanding and hopefully help in the clinical identification and treatment of the disease.

## 2. Materials and Methods

We conducted a thorough literature search with multiple databases, including Pubmed, Embase, Medline, SCIE and CNKI, from 1 January 1991 to 30 June 2022. Our core search keywords for the case literatures related to primary pulmonary NC consisted of the nuclear protein in testis, NUT carcinoma and midline carcinoma combined with the specific term primary pulmonary carcinoma. The relevant journals, bibliographies and reviews were manually searched for additional articles. The search had no language restriction.

Inclusion criteria: (1) real clinical case reports and (2) primary pulmonary NC clearly diagnosed by pathology. Exclusion criteria: (1) duplicate literature, (2) basic experimental research, (3) obvious errors in the data, (4) metastatic pulmonary NC and (5) other tumors besides the lungs.

The search identified 120 references in total. After screening, 35 articles were deemed potentially eligible, and a detailed review was conducted. These original papers were read carefully, the relevant studies were reviewed and information was extracted and compiled from the literature. We extracted the information on the patients' general conditions (including gender, age and smoking history); symptoms; imaging features (lung tumor location and size, lymph nodes and metastases) at initial presentation and pathological characteristics (including histology and immunohistochemistry), as well as the treatment and the final outcome. The statistical analysis was carried out after that.

## 3. Results

### 3.1. Clinical Features

This study evaluated 55 patients (36 men and 19 women) among the 35 articles [3–37] with a mean age of 41.4 years (range, 6–82). The male to female ratio was approximately 1.89 to 1. There were 17 patients aged ranging from 0 to 30 years old, 26 patients aged 31–60 years old and 12 patients older than 60 years old, M (P25, P75) = 36 (26, 54) (M, median; P25, lower quartile; P75, upper quartile). This indicates that primary pulmonary NC occurs not only in young patients but also in older individuals and is more prevalent in men than in women.

Smoking history was not mentioned in 13 cases. Some (*n* =14) patients (33.3% 14/42) explicitly indicated had a history of smoking, while 28 patients (66.7% 28/42) did not, so the ratio was 1:2. A total of 44 cases specifically described the symptoms of primary pulmonary NC patients during their first visit, of which 28 cases reported as a frequent symptom of cough (63.6% 28/44), 13 cases of dyspnea (29.5% 13/44), 8 cases of chest pain (18.2% 8/44), 8 cases of hemoptysis (18.2% 8/44) and 7 cases of wheezing (15.9% 7/44). The remaining symptoms reported were: fever (*n* = 6), pain (except chest pain) (*n* = 6), chest tightness (*n* = 4), expectoration (*n* = 4), weight loss (*n* = 4), debilitation (*n* = 3), neck discomfort (*n* = 2), anorexia (*n* = 2), night sweat (*n* = 2) and hoarseness (*n* = 1). Statistics revealed that the proportion of patients without a smoking history was significantly higher than those with a smoking history. Cough, dyspnea, chest pain and hemoptysis were taken as the main clinical manifestations during the first visit. The details are in Table 1.

### 3.2. Imaging Examination Findings

The pertinent characteristics during the patient's initial imaging (CT or enhanced CT) were mentioned in 51 cases. The majority of imaging was reported with large, irregular soft tissue masses in the lungs and partial involvement of the pleura.

Forty cases mentioned the exact size of the primary lesion, and the lesion was larger than 3 cm in thirty-four cases (85% 34/40); among them, the largest mass was

$10 \times 6.4 \times 12.7$ cm. The maximum diameter of the pulmonary occupancy measured during imaging ranged from 1.4 to 12.7 cm, with a mean diameter of 6.55 cm.

Of the 51 cases mentioned sites of tumor invasion, 22 of left lung invasion (43.1% 22/51), 27 of right lung invasion (52.9% 27/51) and 2 of trachea invasion (3.9% 2/51) were reported. Most (*n* =43) cases illustrated the specific location of the lesion invading in the lung: 20 lower lobe invasion (46.5% 20/43) (10 left, 10 right), 12 upper lobe invasion (27.9% 12/43) (5 left, 7 right), 2 right middle lobe invasion (4.7% 2/43) and 7 hilar region invasion (16.3% 7/43) (2 left, 5 right) (Table 2). Some imaging findings indicated pleural effusion (*n* = 13), compression atelectasis (*n* = 5) or the foci of necrosis (*n* = 2).

**Table 1.** Clinical features.

| Sex | Male (*n*) 36/55 | Female (*n*) 19/55 | Male:Female 1.89:1 | | |
|---|---|---|---|---|---|
| Age | 0–30 (*n*) 17/55 | 30–60 (*n*) 26/55 | >60 (*n*) 12/55 | Mean (years old) 41.4 | Median (years old) 36 |
| Smoking history | Yes (*n*) 14/42 | No (*n*) 28/42 | Yes:No 1:2 | | |
| Symptom | Cough (*n*) 28/44 | Dyspnea (*n*) 13/44 | Chest pain (*n*) 8/44 | Hemoptysis (*n*) 8/44 | Wheezing (*n*) 7/44 |

**Table 2.** Lung lesion imaging information.

| Lesion Location | | *n* < 3 | *n* ≥ 3, < 6 | *n* ≥ 6, < 9 | *n* ≥ 9 | NA | Total (*n*) |
|---|---|---|---|---|---|---|---|
| Left Lung | LUL * | 1 | 0 | 1 | 1 | 2 | 5 |
| | LLL | 0 | 5 | 3 | 0 | 2 | 10 |
| | LH | 0 | 1 | 0 | 1 | 0 | 2 |
| | NA | 0 | 0 | 1 | 1 | 3 | 5 |
| Right Lung | RUL | 1 | 2 | 2 | 0 | 2 | 7 |
| | RML | 0 | 1 | 1 | 0 | 0 | 2 |
| | RLL | 1 | 2 | 3 | 2 | 2 | 10 |
| | RH | 1 | 1 | 2 | 1 | 0 | 5 |
| | NA | 0 | 0 | 3 | 0 | 0 | 3 |
| Trachea | | 2 | 0 | 0 | 0 | 0 | 2 |
| Total (*n*) | | 6 | 12 | 16 | 6 | 11 | 51 |

* NA, not available; RLL, right lower lobe; RML, right middle lobe; RUL, right upper lobe; LUL, left upper lobe; LLL, left lower lobe; RH, right hilar; LH, left hilar.

The situations of the lymph nodes and remote metastasis were mentioned in 37 and 39 cases, respectively. Thirty-four cases detected the regional infiltrating lymph nodes (91.9% 34/37). In 29 cases, the exact location of the lymph node metastases was indicated: 23 were mentioned with mediastinal lymph node metastasis (79.3% 23/29) (6 ipsilateral, 4 bilateral); 14 with hilar lymph node metastasis (48.3% 14/29) (9 ipsilateral, 1 contralateral) and 6 with supraclavicular lymph node metastasis (20.7% 6/29) (2 contralateral, 2 ipsilateral and 1 bilateral). Infraclavicular (*n* = 1), subcarinal (*n* = 3), trachea (*n* = 3), chest wall (*n* = 1) and axillary (*n* = 2) lymph node metastases were also mentioned.

Some (21) cases were identified with distantly metastasis (53.8% 21/39): 15 with bone metastasis (71.4% 15/21), 5 with liver metastasis (23.8% 5/21) and 3 with adrenal metastasis (14.3% 3/21). Other distant metastases in the lung (*n* = 2), ovary (*n* = 1), breast (*n* = 1), retroperitoneum (*n* = 1), brain (*n* = 1), pericardium (*n* = 1) and esophagus (*n* = 1) were also mentioned.

According to the statistical analysis, the primary lesions of pulmonary NC are usually large and mostly located in the lower lobe of the left or right lung (the right lung accounts for a slightly larger proportion), which may be accompanied by pleural effusion, compression atelectasis or pulmonary necrosis. The majority of patients would have lymph node

metastases at the initial imaging examination, primarily in the mediastinum, ipsilateral hilum and supraclavicular regions. Distant metastases also occurred in some cases, most of which were bone, liver and adrenal metastases. (Detailed in Supplementary Table S1.)

*3.3. Pathological Characteristics*

Among all the literatures we reviewed, 47 cases were described in detail regarding their histological features of the primary or metastatic lesions at the surgical specimens or fine needle aspiration (FNA) samples. In summary, sheets of undifferentiated or poorly differentiated (93.75% 30/32) cells with squamous differentiation (*n* = 15) and abrupt keratinization (*n* = 13) were typically observed (32 cases mentioned the degree of cell differentiation). The cytological morphological features included small-to-intermediate-sized (*n* = 7) elliptical cancer cells, with irregular nuclei in a nested arrangement, and overlapped nuclei (*n* = 2) could be seen in a minority of the cases. Occasionally, naked nuclei (*n* = 3) were seen around clusters and debris. Numerous tumor cells (*n* = 25) possessed prominent nucleoli (*n* = 25), scanty cytoplasm (*n* = 7) and a high nuclear-to-cytoplasmic (N/C) ratio (*n* = 4). The cells were predominantly scattered in a necrotic background (*n* = 15), with a nest-like distribution (*n* = 11) in which a distinct neutrophil infiltration (*n* = 8) was present. Apoptosis and identifiable mitotic figures (*n* = 12) could be found in some tumor cells.

Immunohistochemistry (IHC) showed that all specimens were positive for NUT, except for eight cases not mentioned. Most (49) cases specifically described the results of IHC staining: 30 positive for P63 (61.2% 30/49), 16 positive for P40 (32.7% 16/49), 6 positive for TTF-1 (12.2% 6/49) and 5 positive for CD56 (10.2% 5/49). In addition, other epithelial markers (CK5/6, CK7, AE1/AE3, CAM5.2, EMA, etc.) (69.4% 34/49) and neuroendocrine markers (Syn, NSE, CgA, etc.) (20.4% 10/49), as well as other markers such as P53, CD99, LCA, CD138, Vimentin, BCL-2 and HHF-35, can also be discovered as positive. Nineteen cases were positive for Ki-67 (range 25–80%), of which eighteen cases were reported above 30% (94.7% 18/19), demonstrating a generally high but variable proliferation index, as measured by Ki-67 (Figure 1).

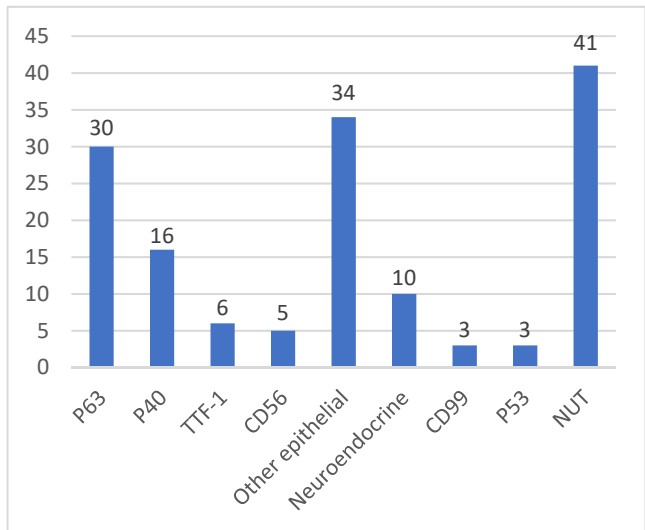

**Figure 1.** The mainly positive signals of IHC.

In 36 cases, modern molecular biotechnology was used to detect the gene fusion, including fluorescence in situ hybridization (FISH) (*n* = 19), next-generation sequencing (NGS) (*n* = 8), reverse transcriptase-polymerase chain reaction (RT-PCR) (*n* = 3), comprehensive genomic profiling (CGP) (*n* = 3) and other diagnosis approaches (*n* = 5). The majority of these cases manifested a tumor with chromosomal translocation, which forms the BRD4-NUT fusion oncogene (*n* = 16), BRD3-NUT (*n* = 4), NSD3-NUT (*n* = 3), NUT-variant (*n* = 1),

CHRM5-NUTM1 (*n* = 1) and other NUTM1 rearrangements (*n* = 5). DDR2, CSF1R, DAXX, RUNX1 somatic mutations, ATXN3 ZNF429 mutations and EGFR exon19 deletion were also identified by NGS. Due to the rarity of NC, these methods are not routinely used in all cancer patients, resulting in the misdiagnosis or delayed identification of this disease. (Detailed in Supplementary Table S2.)

*3.4. Treatments and Outcomes*

Forty cases documented the OS since diagnosis. The median OS in patients with primary pulmonary NC was 4.4 months (range 1 day–26.7 months), and the mean OS was 6.21 months.

Fifty cases described the complete treatment course. Five patients were untreated with just a mean OS of 1.3 months. Eleven patients had surgical resection of the primary lesion (22% 11/50), of which seven patients operating alone had a mean OS of 3.54 months. For the other four patients who received neoadjuvant therapy and postoperative adjuvant chemoradiotherapy, three of them were still alive at the time of reporting, and one of them had an 8-month OS. Patients receiving neoadjuvant therapy or postoperative adjuvant chemoradiotherapy demonstrated an obvious improvement in OS, whereas surgery alone could only have a limited benefit.

Chemotherapy (platinum-based regimen constituted the majority) was chosen by 33 of the unoperated patients (66% 33/50), with a mean OS of 7.32 months, longer than the mean OS of untreated and surgically only patients. Of these, 16 patients were treated with chemotherapy alone, of whom two were alive at the time of reporting, and one did not record the survival time, leaving a mean OS of 5.58 months. Eight patients opted for chemoradiotherapy (applied concurrent chemoradiotherapy or sequential chemoradiotherapy as the first-line therapy), with a mean OS of 13.2 months for the six patients who recorded the survival time, and four of them reported an OS more than 12 months (12.9, 16, 18 and 26.7 months, respectively). On the basis of chemotherapy alone, immunotherapy was administered to three cases: the OS of one case was unclear, and the remaining two cases reported OS of 3 and 8 months, respectively. Seven cases applied immunotherapy in the posterior lines. As most immunotherapies were coupled with other treatments, their efficacy could not be determined with certainty. Six patients were treated with bromodomain and extra-terminal motif inhibitors (BETi) or histone deacetylase inhibitors (HDACi) for the posterior-line treatment, with a mean OS of 7.45 months. There was no significant therapeutic benefit found in this therapeutic option, which may due to the misdiagnosis of most patients and the advanced disease at the time of the confirmed diagnosis. Patients with primary pulmonary NC have a clinical response to chemotherapy, especially combined with radiotherapy, suggesting that the tumor is sensitive to radiotherapy. The effect of a combination with immunotherapy is uncertain. The details are in Table 3.

**Table 3.** Treatments and outcomes.

| Treatment | Recorded/Total (*n*) | Mean OS (Months) |
| --- | --- | --- |
| Untreated | 5/5 | 1.3 |
| Surgery only | 11/11 | 3.54 |
| Surgery combined with adjuvant therapy | 1/4 | (8) * |
| Chemotherapy only | 13/16 | 5.58 |
| Chemoradiotherapy | 6/8 | 13.2 |
| Chemotherapy combined with immunotherapy | 2/3 | 5.5 |

* Only one case in this group with an OS of 8 months.

The TP regimen (Taxanes + Platinum) was chosen as the most common chemotherapy (30.3% 10/33), followed by the EP regimen (Etoposide + Platinum) (18.2% 6/33), DP (Docetaxel + Platinum), AP (Pemetrexed + Platinum), etc. For the second-line and third-line treatments, the regimen included platinum-containing chemotherapy; single-agent chemotherapy; dual or single-agent immunotherapy (Pembrolizumab, Nivolumab, Bevacizumab, Atezolizumab, etc.) and targeted therapy (Gefitinib, Apatinib or Anlotinib);

palliative radiotherapy was also chosen, but the effect was moderate. Only 4 of the 50 patients were treated to the fourth line; all of them had received chemotherapy, radiotherapy, immunotherapy and targeted therapy, and all four had an OS of longer than 12 months.

Some (*n* = 23) cases were misdiagnosed initially and were diagnosed with further examination. Thirteen cases were misdiagnosed as non-small cell lung cancer (NSCLC) (13/23) at the early diagnose, most of which were misdiagnosed as lung squamous cell carcinoma (SCC). Moreover, some have been incorrectly diagnosed with small cell lung cancer (SCLC), neuroendocrine tumor, blood disease, sarcoma and other diseases. (Detailed in Supplementary Table S3.)

## 4. Discussion

The rare descriptions of primary pulmonary NC may be attributable to its poor characterization, which makes it difficult to identify from other tumors, and its rapid progressive course. Consequently, a summary of this tumor's characteristics and therapies is required. In this study, we sum up the clinical and pathological characteristics of 55 patients with primary pulmonary NC and investigate the efficacy of various treatment strategies and prognosis.

The tumors mostly occurred in middle young-aged men (median age, 36; the male to female ratio was 1.89:1) who generally had no smoking history and presented mainly with symptoms of cough, dyspnea, chest pain and hemoptysis. The initial imaging examination showed large and irregular lesions of the lungs, primarily located in the lower lobe of the left or right lung, partially involving the pleura. The majority of them had metastatic lymph nodes in the mediastinum, ipsilateral hilum or supraclavicular regions. Over fifty percent of patients had distant metastases of the bone, liver or adrenal glands. A histological examination of NC usually demonstrated that the tumor cells displayed a focal squamous differentiation with abrupt keratinization, with abundant necrosis and numerous neutrophils within the background, and brisk mitotic rates could also be found. The immunohistochemistry for NUT revealed diffuse nuclear speckles in all the tumors tested. We determined that the mean OS in patients with primary lung NC was 6.21 months, with a median OS of 4.4 months. There is no standard management for primary pulmonary NC. The OS would be prolonged by concurrent chemoradiotherapy or sequential chemoradiotherapy, which plays a significant role in treating primary pulmonary NC, followed by surgery combined with neoadjuvant therapy or adjuvant chemoradiotherapy. Surgery alone or chemotherapy alone also lengthens patients' survival periods to a certain degree. However, NC is difficult to be early clinically identified, and surgical inaccessibility frequently results in a dismal OS for patients with advanced disease. The clinical therapeutic effect of immunotherapy is unknown, which is mainly used as posterior-line therapy. There has been no uniform standard chemotherapy regimen until now, but platinum-containing dual drugs (mostly TP) are commonly applied. Anthracycline, cyclophosphamide (CTX), Ifosfamide (IFO) and other chemotherapeutic agents are also used. HDACi and BETi could not show excellent efficacy in our study, which may be related to the easy misdiagnose and the advanced stage at the time of diagnosis.

The poorly differentiated or undifferentiated nature of the tumor leads to difficult histomorphological identification and a wide range of differential diagnoses. Primary pulmonary NC expresses nuclear markers of squamous differentiation (P63 and P40) and keratins invariably (particularly high molecular weight keratins). Markers such as P63 and P40 are atypical for NC and are generally considered a poorly differentiated type of squamous cell carcinoma. TTF-1 is typically negative but may be focally expressed in a subset of pulmonary NC that can be misidentified for lung adenocarcinoma. However, extensive p63 and p40 expression with focal TTF-1 in the same tumor cell population is extremely rare among lung carcinomas, and NC should be considered [38]. In addition, immunoreactivity with TTF-1 and neuroendocrine markers (Syn, CgA, etc.) will support the diagnosis of small cell carcinoma. The expression of CD99 can lead to the misdiagnosis of Ewing's sarcoma. The expression of LCA and CD34 may be misdiagnosed as lymphoma

and acute leukemia, respectively. Missing the diagnosis can result in possible fatalities. In 2003, French et al. [39] revealed that the t (15–19) translocation observed in these aggressive malignancies occurs in an oncogenic fusion gene between the genes BRD4 and NUT. To date, NUT rearrangements have become the diagnostic hallmark of NC. Recent studies [40] have demonstrated that C52B1 (a NUT monoclonal antibody) has 100% specificity and 87% sensitivity for the diagnosis of NC; therefore, the presence of a diffuse positive NUT protein is reliable evidence for the diagnosis of NC. However, if NUT immunostaining is negative, the diagnosis of NC should not be ruled out, and NUT rearrangement should be conducted by molecular analysis techniques such as FISH or NGS.

NC is defined by the presence of BRD-NUT fusion oncogenes, with the BRD4-NUTM1 gene fusion being the most common. These fusion genes contribute to the aggressive phenotype by promoting cell growth and inhibiting differentiation via aberrant histone acetylation and MYC activation. BETi has been proven to antagonize this process [41] and has shown definitive therapeutic responses in the cases of NC at other sites [42]. HDACi has also been reported to be beneficial in treating NC [43]. Primary pulmonary NC is a disease with a dismal prognosis for which conventional treatments appear to an unsatisfactory effect. BETi and HDACi have emerged as two promising targeted medicines; though their development is still in process, it is anticipated that they will assist NC patients in the future.

Initially, the disease was commonly believed to only occur in young patients, but it is clear now that no one of any gender or age can be excluded. In the presence of abnormal aggressiveness in the clinical process, such as rapid enlargement of the mass and the appearance of lymph nodes or distant metastases in a short period of time, NC should be taken into account even in older women who smoke heavily. The survival period of patients with this disease is extremely short, which is mostly related to the difficulty in identifying NC, but it may also reflect a relative delay in the onset of clinical symptoms of primary NC. In addition, we found that, unlike the common types of lung cancers, including squamous cell carcinoma and adenocarcinoma, there were not many cases of NC metastasis in the brain, possibly because the patients with NC did not have a long enough survival period to develop brain metastases.

Rare tumors will always bring challenges to clinical practices. It is not difficult to tell from the existing literature that the first-line treatment for some individuals with NC is for the treatment of diseases that have been misdiagnosed. Obtaining a prompt and accurate diagnosis is a major obstacle for physicians but vitally crucial for patients. However, tumor molecular detecting and gene sequencing technology are in ascendance; the timely review of these findings by a clinician with related expertise could perfect the clinical diagnosis. A definitive diagnosis will afford a greater possibility to attempt various therapy regimens, including molecularly targeted clinical trials before the therapeutic window closes. It is performed by IHC or modern molecular biotechnology. The greatest challenge is not how to diagnose NC but to determine when to include NC in the differential diagnosis and perform diagnostic molecular tests on NUT-specific antibodies. Both clinical features, imaging and pathological characteristics, should be fully exploited to consider this malignant tumor. In particular, when differentially diagnosing as undifferentiated or poorly differentiated carcinoma of the chest in nonsmokers, clinicians should consider NC.

## 5. Conclusions

In conclusion, we demonstrated that it still has the characteristic diagnostic features of primary pulmonary NC, although it is often misdiagnosed. Typical patients of the disease are relatively young and nonsmoking, tend to have a large lesion in a unilateral lung and are occasionally accompanied by pleural effusion or atelectasis. The tumor cell population often shows p63, p40 and keratin positive from a biopsy. Primary pulmonary NC is confirmed by NUT IHC or molecular examination. The disease is sensitive to concurrent chemoradiotherapy or sequential chemoradiotherapy, but the prognosis is still unsatisfactory. Clinicians should encourage their patients to actively participate in appropriate clinical trials.

**Supplementary Materials:** The following supporting information can be downloaded at https://www.mdpi.com/article/10.3390/curroncol29100536/s1: Table S1. Clinical features of fifty-five patients with primary pulmonary NUT midline carcinoma. Table S2. Pathological features of fifty-five patients with primary pulmonary NUT midline carcinoma. Table S3. Treatments and outcomes.

**Author Contributions:** J.Y.: conceptualization, formal analysis, methodology and writing—original draft. Z.X.: writing—review and editing. Y.G.: supervision. All authors have read and agreed to the published version of the manuscript.

**Funding:** This work is supported by the National Famous Traditional Chinese Medicine Expert Inheritance Studio; the National Natural Science Foundation of China (Grant No. 81973805); Zhejiang Provincial TCM Science and Technology Project (Grant No. 2015ZA088); Zhejiang Provincial Project for the key discipline of traditional Chinese Medicine (Yong GUO, no. 2017-XK-A09, http://www.zjwjw.gov.cn/, accessed on 20 July 2022).

**Conflicts of Interest:** The authors declare no conflict of interest.

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
