# Peer review of "Diagnosis, Treatment and Prognosis of Primary Pulmonary NUT Carcinoma: A Literature Review"

_curroncol, doi:10.3390/curroncol29100536_

Round 1
Reviewer 1 Report
Jiaqian Yuan and Yong Guo present the manuscript entitled “Diagnosis, Treatment And Prognosis Of Primary Pulmonary NUT Carcinoma: A literature review”.
Major Points:
1. Although the manuscript clearly indicates that its oriented to review the Diagnosis, Treatment And Prognosis Of Primary Pulmonary 2NUT Carcinoma, molecular features of this rare type of cancer should be very informative for non-experts readers. Please add a new section reviewing the genetics/ genomics and molecular biology of 2NUT carcinoma.
2. In general, there are many typos and orthographical errors and the English should be thoroughly revised.
3. The authors should discuss, what is the main relevance of this review?
Minor comments:
Introduction
Page 1, line 30 Replace “testis(NUT)” with “testis (NUT)”.
Page 1, line 30 Replace “its” with “it´s”.
The authors should rewrite the introduction as it is very brief and has several typographical errors.
Results
Authors should include a table with clinical characteristics.
Page 2, line 81. Delete the extra period.
Page 2, line 75. Add a space between the word and the opening bracket.
Discussion
The authors should more deeply discuss the results.
Reviewer 2 Report
The work of the authors is appreciated, as the topic seems to be unique and relevant. The representation of the information regarding NUT Carcinoma is good. I would like to share some suggestions for the improvement of the manuscript.
Result:
*Explain the meaning of "M (P25, P75)=36 (26,54)" in section 3.1
Discussion:
*Define unusual abbreviations e.g. CTX, IFO.
*It will be better if the author adds a conclusive sentence about NC disease characteristics, diagnosis, and management.
Round 2
Reviewer 1 Report
Authors have replied all my concerns, thus I strongly suggest to accept the manuscript for publication in its actual form.